# Work-Related Stress, Physio-Pathological Mechanisms, and the Influence of Environmental Genetic Factors

**DOI:** 10.3390/ijerph16204031

**Published:** 2019-10-21

**Authors:** Emanuele Cannizzaro, Tiziana Ramaci, Luigi Cirrincione, Fulvio Plescia

**Affiliations:** 1Department of Health Promotion Sciences Maternal and Infantile Care, Internal Medicine and Medical Specialities “Giuseppe D’Alessandro”, University of Palermo, via del Vespro 133, 90127 Palermo, Italy; luigicirrincione@gmail.com (L.C.); fulvio.plescia@unipa.it (F.P.); 2Faculty of Human and Social Sciences, Kore University of Enna, 94100 Enna, Italy; tiziana.ramaci@unikore.it

**Keywords:** work-related stress, environment, genetic factors, stress

## Abstract

Work-related stress is a growing health problem in modern society. The stress response is characterized by numerous neurochemicals, neuroendocrine and immune modifications that involve various neurological systems and circuits, and regulation of the gene expression of the different receptors. In this regard, a lot of research has focused the attention on the role played by the environment in influencing gene expression, which in turn can control the stress response. In particular, genetic factors can moderate the sensitivities of specific types of neural cells or circuits mediating the imprinting of the environment on different biological systems. In this current review, we wish to analyze systematic reviews and recent experimental research on the physio-pathological mechanisms that underline stress-related responses. In particular, we analyze the relationship between genetic and epigenetic factors in the stress response.

## 1. Introduction

In the professional world, alongside specific risk agents responsible for occupational diseases, there are other factors, such as stress conditions, capable of creating non-specific occupational diseases with more or less serious consequences on the biological, physical, psychological, and social level of the individual [1,2,3]. Stress related to work is one of the most frequent causes of occupational diseases, including cancer. The European Agency for Safety and Health at Work has adopted the following definition for stress: “Work-related stress is experienced when requests from the work environment exceed the individual’s ability to deal with this request”. 

Although the importance of stress as a pathogenetic factor of various psychosomatic illnesses has been recognized ever since the first half of the last century, it is only in these last decades that the deepening of knowledge in the fields of neurochemistry, neuroendocrinology, and immunology has allowed us to understand and interpret, in large, the variations of the specific mechanisms that are at the basis of pathologies induced by an excess of adverse stimuli [4,5].

As complex as they may be, the responses to stress in humans are the expression of an integrated and genetically controlled biological program [6,7]. The motivations able to invoke an alarmed reaction are part of the individual’s daily life, and the possibility that the person can reduce or cancel the negative consequences depends on their ability to adapt. Stressors that are completely similar can induce quantitatively and qualitatively different responses in accordance with the personality and the experiences of the individual, their biorhythms, and the characteristics of the stressors. Characteristics such as regularity, predictability, avoidability, duration, and intensity, as well as various environmental factors such as the light/dark cycle, temperature, humidity, noise, ionization of the atmosphere, and the frequencies of magnetic fields can all influence the stress response in different ways [8,9,10,11,12,13,14,15].

In recent years, within this high-technological progress society, the types of stress to which a subject is exposed are not predominantly of a physical nature. The most frequent stressors to manage and to which subjects are exposed are those of a psychosocial nature, favored by a widespread social organization concentrated on pragmatism and industrialization, and characterized by intense emigration and rapid changes in the socio-economic status of the individual, in addition to a progressive weakening of the family structures and the supports provided by the society [16,17,18,19].

In this regard, it seems that the European economic crisis, which began in 2008, could play a prominent role in an individual’s probability of contracting stress-related disease [20,21,22,23,24,25]. In particular, it has been reported that financial crisis, loss of work, and reductions in salary could significantly increase the frequency of mental health disorders and the consumption of substances of abuse. Furthermore, in times of crisis, health outcomes and the risk of health-related financial hardship may be affected by changes in the resources available for health systems (e.g., financial and human resources, drugs and medical devices, running costs and infrastructure), by changes in living conditions, lifestyles, and consumer behaviors, as well as by changes in social norms and values. On the other hand, several critical studies demonstrated that financial crises seemed to be linked to increased work-related stress, and in some cases, to the development of mental illness [26,27,28,29,30,31,32,33,34]. An elegant review by Mucci and colleagues [35] analyzed results about the economic crisis and physical health, showing that financial crisis was an important stressor that was able to promote negative effects on the health of workers and the general population. 

The homeostatic adaptations to stress are essentially regulated by the central nervous system (CNS), the neuroendocrine system (NES), and the immune system (IS) [1,36,37,38,39,40,41]. These closely connected systems allow us to perceive, process, and transform the stimuli into messages for the various effector organs. The brain is made up of a series of synaptic connections wired between neurons, comparable to thousands of servers that connect millions and millions of computers. The neuron that transmits the signal releases one or more neurotransmitters and neuromodulators into the synaptic space, and these selectively interact with receptors on the surface of the neuron to which the information is to be transmitted. The interaction triggers a cascade of specific biochemical events that are responsible for the appearance of various biological and behavioral effects. This feature allows the different brain areas to interact synchronously to perform complex tasks. Stress-specific variations are determined in the release of neurotransmitters and neuromodulators that, for a short or long time, will influence the responses of the various organs and behaviors, such as emotional, cognitive, alimentary, and sexual [42,43,44,45,46,47]. 

The objective of this qualitative systematic review is to examine and interpret the evidence from different studies on neuroendocrine and genetic factors that underline and predict vulnerability to work-related stress responses.

## 2. Materials and Methods 

The author’s search targeted evidence-based guidelines, evidence-based summaries, systematic reviews, and recent experimental research on the physio-pathological mechanisms that underline stress-related responses. The keywords used were “work-related stress”, or “neuroendocrine stress responses”, or “stress and immune system”, or “cytockine and hypothalamic-pituitary-adrenal (HPA) axis”, or “genetic and epigenetic programming of stress responses”. Through this strategy, we identified more than 1000 papers using two primary sources for identifying relevant information: PubMed and SCOPUS (last accessed via PubMed and SCOPUS on September 2019).

## 3. Neuroendocrine Control of the Stress Response

The stress response is characterized by numerous neurochemicals, neuroendocrine and immune modifications that involve various neurological systems and circuits, as well as regulation of the gene expression of the different receptors. The main brain structures involved in stress responses include the prefrontal cortex, the amygdala, the hippocampal septum system, and the adrenergic nuclei of the brain stem, including the nuclei coeruleus, paravertebral, cuneiform, and the dorsal nucleus of the raphe [41,48,49,50,51].

Corticotropic releasing hormone (CRH) and adrenalin (AD) are the main brain mediators that coordinate stress responses. In particular, the CRH response is mediated by two receptors, CRH-R1 and CRH-R2. CRH-R1 receptors are involved in processing sensory information and motor control, while CRH-R2 receptors regulate emotional, affective, and cognitive behavior [52,53,54,55]. A typical neuroendocrine stress response involves the immediate release of CRH and vasopressin from the hypothalamus that, by mutual reinforcement, stimulates the pituitary to secrete the adrenocorticotropic hormone (ACTH), which activates the secretion of cortisol (COR) from the adrenal gland [56,57].

The effects of COR are mediated by two subtypes of receptors: The mineralcorticoid receptors (MRs), which are continually occupied by COR during the day and exert a tonic inhibition on the activity of the HPA axis, and the glucocorticoid receptors (GRs), which are activated only when the COR levels are elevated, as happens during stress or the morning peak [58,59,60,61,62,63]. It has been observed that a down-regulation of the MRs determines a serious deficit of learning and memory, while a down-regulation of the GRs allows individuals to still be able to learn and remember past experiences [39,64,65,66,67,68]. A greater release of glutamic acid also contributes to the negative effects of stressors, such as a reduction in serotonin as well as in the levels of Brain-Derived Neurotrophic Factor (BDNF), a peptide that regulates the proliferation and differentiation of synapses and the survival of neurons, which is expressed in numerous brain structures [69,70,71]. Increased levels of AD in acute stress facilitate the formation of memories associated with intense emotions, while in chronic stress, elevated COR levels are maintained for a long time with consequent disturbances of emotional, affective, and cognitive behavior [72].

The locus ceruleus is a critical component of the brain’s vigilance system as it controls the subject’s state of alert. A rapid activation of the locus ceruleus–AD system contributes to the processing of somatic adaptation responses to stress, such as increased blood pressure and heart rate, erection pacing, and mydriasis, as well as activation of metabolic processes [73].

The HPA axis and the locus cereuleus–AD system differ in their temporal responses. The locus ceruleus response is fast and runs out quickly, while the HPA axis response starts immediately and lasts longer. There is an increase in the blood levels of CRH and AD through a negative feedback mechanism during post-stress periods, which reduces the production and secretion of CRH and AD. CRH, in addition to controlling the response of the HPA axis, also acts as a neurotransmitter in various extra hypothalamic circuits in order to effectively integrate multiple brain responses to stressors. In addition to activating the alert state and increasing motor reflexes and emotional tone, CRH also influences cognitive processes [1,39,73,74,75].

## 4. The Immune System’s Role in the Stress Response

The CNS, peripheral nervous system (PNS), and IS are part of a completely integrated biological circuit, and their signals are used both for the exchange of information between the elements of the same system and for communication between the three systems. The CNS modifies its responses through both self-regulation mechanisms and signals coming from the NES and the IS. The existence of a bidirectional network of communication between the NES and IS causes the body to respond to non-cognitive stimuli, such as those of an infectious, immune, and neoplastic nature. Changes in the responses of the NES and IS due to stress and the genetic characteristics of the subject can also depend on the meaning attributed to the stressors in relation to the subject’s previous experiences and the presence or absence of social support [76,77,78].

COR and AD, which are released following the activation of the HPA axis and the central sympathetic system, are the main compounds that modulate the IS response [79,80,81]. After an event of acute stress, there is an increase in the blood levels of AD due to activation of the HPA axis which therefore increases the production of specific mediators—the cytokines—by the IS [82,83,84,85]. The cytokines are chemical messengers that flag the presence of danger coming from non-cognitive stimuli to the organism. These cross the blood–brain barrier with an active transport mechanism, facilitating the release of CRH and ACTH from the hypothalamus and the hypophysis [86,87]. Once the stress response is over, the elevated blood levels of COR, through a central counter-regulation mechanism, bring the activity of the IS back to the basal levels. If the stress is prolonged, the repeated activation of the HPA axis and of the central adrenergic system induces a persistent increase in the blood levels of COR and AD, which activates the IS and predisposes the subject to numerous types of pathologies [86,87,88,89].

## 5. Role of Genetic and Epigenetic Factors in the Stress Response

Different studies have reported that epigenetics is a risk factor for the onset of stress-related disorders [90,91]. The possibility that environmental work stresses can cause organic pathologies and behavioral disturbances is inversely proportional to their genetic component [92]. Pathologies with a strong genetic component may manifest themselves in response to mild or moderate stressors, while intense and prolonged stressors are required to induce pathologies characterized by a lower genetic determination. Psychiatric disorders such as schizophrenia can be unveiled by stressors of modest intensity, while a depressive syndrome needs severe stimuli to manifest itself, as it has a minor genetic component. As in the case of post-traumatic stress disorder, if the magnitude of the stressful event is particularly high, a mental illness can also arise in subjects with normal and resilient genomes [93,94]. Two possible mechanisms with which genes and the environment can interact are: "Genetic control of sensitivity to the environment" and "genetic control of exposure to the environment". Genetic control of sensitivity to the environment suggests that genes, at least in part, make individuals more or less vulnerable to the effects of environmental stressors, while genetic control of exposure to the environment implies that genetic factors influence the likelihood of individual selections, such as the necessity to emigrate for work, or exposure to environments with lower risks of stress [95,96,97].

The ontogenesis of the brain, from the first fetal outline to complete development, is characterized by a series of histological and biochemical changes induced by the continuous interaction of genes with the environment. In particular, CRH plays a decisive role in the maturation of the HPA axis, as it can regulate the development of neuronal circuits and connections aimed at stress responses. CRH also interacts with the neurons that synthesize serotonin, GABA, and BDNF, which have trophic effects on the brain during the early periods of ontogeny [98,99,100].

There is vast literature that demonstrates how the quality of the environment has a deep impact on the development of different brain circuits. Most of the knowledge of the early effects of stress on humans comes from retrospective studies of children whose mothers suffered psychological stress during pregnancy. These children showed delayed motor development, reduced attention, excessive reactivity and aggression, mood disturbances, irresponsible social behavior, and in adulthood they became more vulnerable to anxiety and depression disorders, and encountered a greater risk of taking drugs. It has been suggested that these behavioral disorders are mediated, at least in part, by the excessive activation of the mother’s HPA axis. According to these observations, experimental animal studies have shown that intense and persistent prenatal stress induces, from the first days of birth, a dysfunction of the HPA axis with an increase in the release of CRH, ACTH, and cortisol in the young ones, as well as an accentuated response of the sympathetic autonomic system. These effects are accompanied by a loss of neurons and receptors in the brain areas that control stress responses and correlate with an increase in emotionality, depression, minor social interaction, and cognitive impairment [101,102].

In recent years, epigenome studies have shown that modifications in the regulation of gene expression are transferable from one generation to another through the parent’s sperm. “It has been demonstrated that sperm is not meant to only transport 23 chromosomes, but it also carries an epigenetic cargo consisting of methylated DNA, non-coding RNAs, protamines, and histones which are critical for fertilization and programming early embryonic development. While observing multiple animal studies, it resulted that paternal stress before conception was associated with changes in sires’ sperm miRNAs (small molecules that have the function of silencing and degrading specific messenger RNAs, preventing their translation into proteins), decreased HPA axis responses following an acute stressor in the child, and increased expression of glucocorticoid-responsive genes in the brain of the child” [103,104,105,106]. 

There is limited epidemiological data on the effects of human paternal preconception exposure on children’s health. However, in a series of recent studies of human epidemiology among adults born to mothers and father with Holocaust post-traumatic stress disorders (PTSD), this condition was associated with an increased risk of PTSD, lower levels of urinary cortisol, increased glucocorticoid sensitivity, and lower methylation of the GRs gene in the child [107]. 

## 6. Conclusions

In the professional work environment, together with specific risks responsible for occupational diseases, there are non-specific psychosocial and environmental risks that can determine different organic and behavioral disorders. The homeostatic adaptations to stress are regulated by the CNS, NES, and IS, which constitute an integrated biological circuit under the control of genes. COR and the AD, released after the activation of the HPA axis and the central sympathetic system, are the main factors that modulate the stress response. Pathologies with a strong genetic component can be triggered by mild stressors, while those with a modest genetic component require more severe stressors. Brain ontogenesis is characterized by continuous neurobiological changes induced by a persistent interaction of genes with the environment. Retrospective studies in the human field have shown that psychological stress during pregnancy determines emotional, affective, cognitive, and social behavior disorders in the child that correlate with a dysfunction of the HPA axis and with an increased response of the sympathetic autonomic system. 

Recent experimental research on epigenomes has shown that modifications in the regulation of gene expressions are transferable through the parent’s sperm to the child. In particular, animal studies have shown that paternal stress before conception induces modifications in the miRNAs of the sperm to the child which determine a dysfunction of the HPA axis associated with various behavioral disorders. These recent studies demonstrate how, when both parents have the knowledge of the negative consequences of psychosocial stress on the health of their child, it is essential to be able to implement a correct lifestyle before and after conception. Undoubtedly, a decisive contribution to this knowledge must be given by all social figures who are required to guarantee the health and well-being of the worker. 

This qualitative systematic review [108] provides further information about the roles played by homeostatic adaptations to stress that are responsible for the appearances of various biological and behavioral effects. Furthermore, our review, by focusing on the involvement of epigenetic factors as risks for the onset of stress-related disorders, provides new avenues of research regarding the mechanisms with which genes and the environment interact and predispose individuals to the onset of stress-related illness.

Taken together, these observations highlight the demand for further investigations that are able to identify and characterize the specific neuroendocrine and genetic factors that subtend and predict vulnerability to the work-related stress response.

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
