# Peer review of "Work-Related Stress, Physio-Pathological Mechanisms, and the Influence of Environmental Genetic Factors"

_ijerph, 2019, doi:10.3390/ijerph16204031_

Round 1
Reviewer 1 Report
This is a good article which summarizes the topic area very well. However, the methodology and evaluative component is weak. It is not stated what type of review has been conducted, and each review type has specific criteria to fulfil. For example, it is not a systematic review as there are no specific inclusion or exclusion criteria. Furthermore, the research that has been conducted to date is not evaluated. If it is a scoping review, you do not have to evaluate the evidence. The authors must decide what type of review this is, and follow the exact criteria required for that type of review. There are many options (meta analyses, rapid review, umbrella review, systematic review, state of the art review etc). See Grant, M. J., & Booth, A. (2009). A typology of reviews: an analysis of 14 review types and associated methodologies. Health Information & Libraries Journal, 26(2), 91-108.
These are example guiding questions you can use for evaluating your review (in bold some aspects not addressed currently):
Is the review question clearly and explicitly stated?
Were the inclusion criteria appropriate for the review question?
Was the search strategy appropriate?
Were the sources and resources used to search for studies
adequate?
Were the criteria for appraising studies appropriate?
Was the likelihood of publication bias assessed?
Were recommendations for policy and/or practice supported by
the reported data?
Were the specific directives for new research appropriate?
The article also needs to recommend some future directions, and a conclusion is missing. In order for the article to be more than a literature review, it needs to have some element of critique of the research done to date.
Author Response
Following the example guiding questions, that reviewer gave us, we have specify our type of review. Line 83-85
As reviewer suggested, we also have carefully explain future direction and improved the conclusion.

Reviewer 2 Report
The manuscript describes the hypothesis, study, and methods of an original research. The Authors aimed to analyze systematic reviews and recent experimental research on the physio-pathological mechanisms that underline stress related responses, with particular regard to the relationship between genetic and epigenetic factors in stress response. It is a current and interesting theme and there is a need to study this issue.
The language, specifically the grammar, is overall discreet. There is the need to correct some inaccuracies and to improve reading fluency. Abstract and keywords are discreet both in terms of appropriateness of context and the purpose of study. The aim of your research has been sufficiently highlighted.
The introduction is sufficiently well written, with an exhaustive analysis of the literature. The global economic crisis that began in 2008 - whose effects are still ongoing - has had a very important impact on society and psychosocial behaviors. Consequently, I invite you also to consider a brief discussion regarding the potential effects of a global economic crisis context on stress-related responses. I state that there are no specific references to literature, and, especially for this, it would be interesting that you reflect on this aspect. Regarding the dynamics of the economic crisis, you can refer to the following publications:
Mucci N et al. The correlation between stress and economic crisis: a systematic review. Neuropsychiatr Dis Treat 2016; 12:983-993. doi: 10.2147/NDT.S98525.
Parmar D, et al. Health outcomes during the 2008 financial crisis in Europe: systematic literature review. BMJ. 2016 Sep 6;354:i4588. doi: 10.1136/bmj.i4588.
Methods section and Results sections appears of a more than sufficient quality. Finally, I suggest you to use a final paragraph to sum up your findings. Otherwise, you should also carefully explain what is the specific contribution that your findings brings to literature and knowledge in this area.
Check accurately all the quotes in the brackets in the text and in the Reference section. These must strictly comply with the Author's guidelines.
Author Response
RESPONSE 1
Line 57-71. Accordingly, with review suggestion, we have considered and discussed the potential effects of a global economic crisis context on stress-related responses.
RESPONSE 2
Line 229-236. We have carefully explain the specific contribution of our review.
RESPONSE 3
Done

Round 2
Reviewer 1 Report
Thank you for clarifying the type of review. The article will be well received by the academic community.
I have corrected this sentence (grammar)- please change in the final version.
The objective of this qualitative systematic review was to examine and interpret the evidence from different studies on neuroendocrine and genetic factors that underline and predict vulnerability to work-related stress responses.
It may be worth getting someone to do a final grammar check.
Best wishes